# A Comparative Analysis of Selection Pressures Suffered by Mitochondrial Genomes in Two Planthopper Species with Divergent Climate Distributions

**DOI:** 10.3390/ijms242316847

**Published:** 2023-11-28

**Authors:** Kang-Kang Sun, Yi Ding, Lei Chen, Jing-Tao Sun

**Affiliations:** Department of Entomology, Nanjing Agricultural University, Nanjing 210095, China; 2020102089@stu.njau.edu.cn (K.-K.S.); 202202024@stu.njau.edu.cn (Y.D.);

**Keywords:** mitochondrial genome, environmental adaptation, selection pressure

## Abstract

Mitochondrial DNA (mtDNA) has been widely used as a valuable tool in studies related to evolution and population genetics, under the implicit assumption of neutral evolution. However, recent studies suggest that natural selection also plays a significant role in shaping mitochondrial genome evolution, although the specific driving forces remain elusive. In this study, we aimed to investigate whether and how climate influences mitochondrial genome evolution by comparing the selection pressures acting on mitochondrial genomes between two rice planthoppers, *Sogatella furcifera* (Horváth) and *Laodelphax striatellus* (Fallén), which have different climate distributions. We employed the dN/dS method, MK test and Tajima’s D tests for our analysis. Our results showed that the mitochondrial genomes of the two species appear to undergo predominantly purifying selection, consistent with the nearly neutral evolution model. However, we observed varied degrees of purifying selection among the 13 protein-coding genes. Notably, *ND1*, *ND2*, *ND6*, *COIII*, and *ATP8* exhibited significantly stronger purifying selection and greater divergence between the two species compared to the other genes. Additionally, we observed relatively stronger purifying selection in the mitochondrial genomes of *S. furcifera* compared to *L. striatellus*. This difference could be attributed to varying metabolic requirements arising from distinct habitats or other factors that are unclear here. Furthermore, we speculate that mito-nuclear epistatic interactions may play a role in maintaining nonsynonymous polymorphisms, particularly for *COI* and *COII*. Overall, our results shed some light on the influence of climate on mitochondrial genome evolution.

## 1. Introduction

Mitochondrial DNA (mtDNA) has been widely used as a molecular marker in diverse fields related to evolution, spanning phylogenetics, phylogeography, population genetics, DNA barcoding, etc., for ~40 years [1]. This extensive adoption of mtDNA as a research tool is largely attributed to its unique features, which make it highly suitable for evolutionary studies. These distinctive features encompass a high mutation rate, maternal inheritance, the absence of recombination, and a high copy number [2,3]. While the genetic variations of mtDNA have been used under the implicit assumption of neutrality, recent studies suggest that positive selection also plays a critical role in mitogenome evolution, questioning that mtDNA is more than an evolutionary bystander [3,4].

Typically, the animal mitochondrial genome (mitogenome) is circular and ranges from 15–20 kb in length, containing 13 protein-encoding genes (PCGs), two rRNAs, and 22 tRNAs [5]. These genes interact with nuclear genes to carry out essential functions such as oxidative phosphorylation (OXPHOS), cellular Ca^2+^ signaling, apoptosis, cell transport, thermoregulation, immunity, etc. [6]. Given its crucial role, mtDNA is thought to evolve primarily under purifying selection. However, more and more evidence suggests the involvement of positive selection in shaping mtDNA evolution. One compelling line of evidence comes from the relationship between the level of mtDNA diversity and species abundance. Bazin et al. [7], in an exhaustive analysis of the mtDNA diversity of ~3000 animal species, discovered no correlation between mtDNA polymorphism and species abundance. They proposed that the recurrent adaptive evolution of mtDNA might explain this phenomenon. In support of this proposition, several studies have revealed the adaptive evolution of mitogenomes related to environmental factors, diet, and behaviors, etc. [1,8,9,10,11,12]. Despite the growing evidence of mtDNA adaptive evolution, our understanding of the driving forces and underlying mechanisms involved remains limited.

The white-backed planthopper (WBPH), *Sogatella furcifera* (Horváth) [13], and the small brown planthopper (SBPH), *Laodelphax striatellus* (Fallén), are both significant pests of rice, but their geographical distributions differ. In addition to causing direct damage to rice plants by sucking sap, these two species can also act as vectors for various rice viruses. WBPH is predominantly found in tropical/subtropical Asian countries, including Laos, Thailand, Vietnam, and parts of Southern China [14,15]. It cannot overwinter in temperate Asian countries, including Japan, Korea, and Northern China, and migrates annually from subtropical regions (Indochina) to temperate regions (East Asia) [16,17]. In contrast, SBPH primarily occurs in temperate regions, where it can endure cold winters through diapause but has a lower tolerance to high temperatures compared to WBPH [18].

In this study, we sequenced 28 unique complete mitogenomes of WBPH. With the assistance of the previously sequenced mitogenomes of SBPH [9], we compared the selection pressures suffered by mitogenomes between the two rice planthoppers using the dN/dS method, MK tests, and Tajima’s D tests. Our aim is to investigate whether and how climate influences mitochondrial genome evolution.

## 2. Results

### 2.1. WBPH Population Genetic Structure

A 1849 bp sequence (partial *COI* + *tRNA-Leu* + partial *COII*) was successfully amplified and sequenced for 159 WBPH individuals sampled from six localities in southern China (Figure 1A). We characterized 28 haplotypes, with six to ten haplotypes for each population (Figure 1B). The relationship among different haplotypes is shown in Figure 1B. The haplotype network analysis revealed that Hap-1 occupied a central position in the star-shaped network and was present in all six WBPH populations with the highest frequency (59.75%), suggesting that Hap-1 is a potential ancestral haplotype. Hap-6, Hap-9, and Hap-10 were also found in multiple populations. Moreover, each population possessed its own private haplotype, ranging from 3 in the FJ, GZ, and YN populations to 6 in the SC population. We observed moderate to high levels of haplotype diversity (Hd) across the populations. The Hd values ranged from 0.496 in GZ to 0.776 in SC, with an average of 0.609. However, the nucleotide diversity values were extremely low, ranging from 0.037% in GZ to 0.074% in SC, with a mean value of 0.053% (Table 1).

A relatively low level of genetic differentiation was observed among the populations, with pairwise *F*_ST_ values ranging from −0.01748 (GX vs. YN) to 0.09903 (SY vs. FJ) (Figure 2A). FJ population displayed some degree of genetic differentiation from the other populations. Principal coordinate analysis (PCoA) based on *F*_ST_ values revealed a clustering pattern among the six populations, in which they could be roughly divided into three clusters: YN, GX, and GZ formed one cluster, SY and SC formed another, and FJ constituted a separate cluster apart from the other two (Figure 2B). This clustering pattern did not show any correlation with geographical locations, suggesting the absence of an isolation-by-distance effect.

### 2.2. Mitogenome Genetic Diversity and Divergence

To investigate the evolutionary characteristics of the WBPH and SBPH mitogenomes, we analyzed the synonymous (Syn) and non-synonymous (Nsyn) mutations in 13 mitochondrial PCGs. In WBPH mitogenomes, the proportions of Syn and Nsyn substitutions in 13 mitochondrial PCGs ranged from 0% to 0.5% and 0% to 0.09%, respectively. In contrast, the corresponding ranges in SBPH mitogenomes were from 0.5% to 2.9% and from 0% to 0.19%, respectively (Figure 3). Wilcoxon signed-rank tests revealed that WBPH mitogenomes had significantly lower proportions of Syn substitutions compared to SBPH mitogenomes (*p* = 2.441 × 10^−4^, Figure 3A). Regarding Nsyn substitutions, WBPH generally displayed lower proportions than SBPH mitogenomes, except for *COIII*, *ATP6*, and *ATP8*. However, the difference in the proportions of Nsyn substitutions was not statistically significant (*p* = 0.05737, Figure 3B).

The average K2P genetic distances between the two species at each PCG ranged from 0.222 (*COI*) to 0.389 (*ATP8*), with a mean value of 0.284 (Figure 3C). Uneven genetic divergence across the 13 PCGs between the two species was noticed, with *ND2*, *ND6*, and *ATP8* showing relatively higher genetic divergence than the others.

### 2.3. Selection Pressure Comparisons

The mean dN/(dS + constant) ratios of the 13 PCG genes were consistently <1, indicative of the prevalence of purifying selection. The Wilcoxon rank-sum test revealed that six PCGs, namely *ND2*, *ND3*, *ND4L*, *ND5*, *ND6*, *CYTB*, *COI*, *COII*, and *ATP6*, exhibited significantly lower dN/(dS + constant) ratios in WBPH compared to SBPH (Figure 4). This indicates a stronger purification selection acting on these genes in WBPH than in SBPH. Conversely, only *COIII* and *ATP8* displayed significantly lower ratios in SBPH than in WBPH.

The McDonald–Kreitman (MK) test with Jukes and Cantor corrections did not identify any PCG showing significant divergent selection between the two species (Table 2). However, NI values significantly greater than 1 were observed for *CYTB* (NI = 4.189, *p* < 0.001), *COI* (NI = 3.296, *p* < 0.001), and *COII* (NI = 4.55, *p* < 0.001) genes, indicating an excess of amino acid polymorphisms segregating within species.

Tajima’s D tests revealed that all 13 PCGs were subjected to purifying selection in both species, except for *ATP8* in SBPH, where a non-significant positive value was observed. However, only four genes, namely *ND1*, *COI*, *COII,* and *ATP6,* in WBPH displayed significantly negative values, suggesting much stronger purifying selection suffered (Figure 5).

## 3. Discussion

The lack of recombination and the high mutation rate make the mtDNA a convenient molecular marker for reconstructing gene genealogy and inferring population history [2]. However, the mitochondrion is also an important organelle that produces up to 95% of a eukaryotic cell’s energy through oxidative phosphorylation. Variation in mitochondrial PCGs involved in oxidative phosphorylation can directly influence metabolic performance [3,4,9,19]. The mtDNA mutation rate is reported to be 10–17 times higher than nuclear genomes due to oxidative damage caused by reactive oxygen species during OXPHOS [20]. This relatively higher mutation rate also makes the mitochondrial genome an ideal candidate to study adaptive evolution. In this study, we investigated the adaptive evolution of the mitochondrial genome in two rice planthoppers that exhibit obvious divergence in climate distribution using dN/dS and variation frequency-based methods.

Population genetics analyses based on partial mtDNA sequences demonstrated that WBPH possessed a similar population structure to SBPH when analyzed using a roughly equivalent molecular marker [18]. This similarity is characterized by low levels of population differentiation, moderate haplotype diversities, and star-like mtDNA networks, which are likely associated with the long-distance migration behavior of both species. Given their comparable population genetic structure, it is meaningful to compare the differences in the strength of selection acting on the mitochondrial genome between the two species. However, the nucleotide diversity in WBPH populations is nearly three times lower than that in SBPH populations. A comparison of the unique mitochondrial genomes of the two species also showed a similar pattern (Figure 3A,B). These findings suggest a relatively smaller effective population size for WBPH.

Our results generally point to a nearly neutral evolution of the mitochondrial genomes of the two species. The MK test revealed that 7 of the 13 mitochondrial PCGs had NI values below 1, but none of them were statistically significant. Among the remaining 6 PCGs with NI values above 1, only the *CYTB*, *COI*, and *COII* genes achieved significance (Table 2). The MK test assumes that the ratio of nonsynonymous substitutions to synonymous substitutions between species should be equal to the ratio of nonsynonymous to synonymous polymorphisms within species [21]. It has been widely used to detect positive selection signals. NI values below 1 indicate an excess of interspecies divergence of amino acid changing substitutions and a strong influence of positive selection. Given this, our MK test results suggest the mitochondrial genomes of the two species generally follow a pattern of nearly neutral evolution, where variations that are neutral or have only slight deleterious effects can accumulate within species.

NI values greater than 1 indicate an excess of amino acid polymorphisms segregating within species compared to what would be expected under a strictly neutral model. This excess suggests a strong influence of purifying selection in shaping the divergence in intraspecies and interspecies ratios [3]. Similar patterns have been observed in studies of various organisms, including *Drosophila,* humans, murids, and other animal taxa. However, it is important to note that the excess of amino acid polymorphisms segregating within species is not solely attributed to the accumulation of synonymous variations with slight deleterious effects but can also be influenced by balancing selection [3]. In support of this possibility, we also found mito-nuclear interactions in SBPH in our previous work [22]. Specifically, when two mitochondria with four nonsynonymous substitutions in their mitochondrial genomes were placed on opposite nuclear backgrounds, it resulted in decreased fecundity [9]. The genetic mechanisms underlying mito-nuclear epistatic interactions involve various physiological and biochemical processes that rely on the coordination between nuclear genes and mitochondrial genes, such as OXPHOS and transcription/translation of mitochondrial genes [6,23,24]. Mito-nuclear interactions have been documented in flies, copepods, seed beetles, and other organisms [25,26,27], suggesting an important mechanism in the maintenance of mtDNA polymorphisms.

The *COI* and *COII* subunits are directly involved in the electron transfer and proton translocation processes, while *COX3* is thought to have regulatory roles [28]. Nonsynonymous mutations in these two genes may have disproportionate effects on fitness, making them typically the most highly conserved in the mitogenome. Consistent with this, previous studies on mammals and spider mites have found that *COX1* and *COX2* consistently lack nonsynonymous mutations, resulting in relatively lower dN/dS ratios [28,29]. In contrast, in our study, we did not observe an obvious lower dN/dS ratio for the two genes than others in the two species. However, we did observe extreme negative Tajima’s D values for these two genes, with the values of WBPH achieving significance, suggesting the two genes were under much stronger purifying selection. Interestingly, we also observed multiple peaks in the distribution of dN/dS values for these two genes, indicating the presence of several groups of mitochondrial genomes that differ in terms of nonsynonymous mutations. These differences in nonsynonymous mutations among groups of mitochondrial genomes could lead to elevated dN/dS values. Considering the possibility of mito-nuclear epistatic interactions mentioned earlier, we speculate that such interactions probably play a role in maintaining these nonsynonymous mutations. To validate this possibility, it would be necessary to conduct bioassays involving the placement of different mitochondrial genomes differed by nonsynonymous mutations into different nuclear backgrounds.

Although no obvious adaptive selection was observed on the mitochondrial genomes of the two species, we found that the mitochondrial genomes of WBPH underwent relatively stronger purifying selection compared to those of SBPH (Figure 4). A prior study in *Caenorhabditis briggsae* also found a similar pattern, in which the πN/πS ratios were higher in the temperate-clade isolates compared to the tropical-clade isolates [30]. The exact reasons for this phenomenon are still unclear, but we suspect that different metabolic requirements resulting from varying environmental temperatures may play a role in shaping the evolution of the mitochondrial genome. Despite mtDNA being inherited as a closely linked circular molecule, the selection pressures acting on the 13 protein-coding genes were uneven, likely due to different functional constraints among genes. *ND1*, *ND2*, *ND6*, *COIII*, and *ATP8* obviously suffered stronger selection pressures than others. Of note, the protein coded by *ATP8* plays a crucial role in the assembly of F0 and exhibited the strongest signal of adaptive variation. This gene was highly conserved in the two species, with only one nonsynonymous mutation found in 28 WBPH mitochondrial genomes and no nonsynonymous mutations detected in all 83 SBPH mitochondrial genomes analyzed. However, a considerable proportion (35/83) of synonymous mutations was observed in the 83 SBPH ATP8 sequences, leading to positive Tajima’s D values. Nevertheless, this gene displayed significantly higher genetic distances between the two species compared to the other genes, suggesting the involvement of divergent adaptive selection in driving mtDNA divergence between the two species.

## 4. Materials and Methods

### 4.1. Sample Collection and DNA Extraction

From July to August 2010, a total of 159 adult females of WBPH were collected from 6 locations in southern China. At each location, adult individuals were randomly sampled from rice plants within a 10 × 10 m area. All collected samples were promptly preserved in absolute ethanol and then stored at −20 °C for genome extraction. The entire body of each specimen was used for total genomic DNA extraction, following the manufacturer’s protocol for the Wizard^®^ SV Genomic DNA Purification Kit (Promega, Madison, WI, USA).

### 4.2. Mitochondrial Haplotype Characterization

Unique mitogenomes were first identified by amplifying and sequencing a specific segment of the mtDNA sequence (partial *COI* + *tRNA-Leu* + partial *COII*, 1889 bp) for each individual. The resulting sequences were used to classify individuals into distinct haplotypes. PCR amplification of the target region was performed using the primers in Appendix A. A total of 28 haplotypes were identified within the WBPH populations. To visualize the relationship between these haplotypes, a median-joining network was constructed using Network 4 [31]. For subsequent mitogenome sequencing, one individual representing each haplotype was randomly selected.

### 4.3. Mitogenome Sequencing and Annotations

The complete mitogenome of WBPH was obtained by amplifying and sequencing two overlapping regions using two pairs of primers (Appendix A), which were designed based on a previously reported mitogenome sequence [32]. Each PCR amplification was performed in a total volume of 50 μL PCR mix, containing 1× Gflex PCR Buffer, 1.25 U TKS Gflex DNA polymerase (TaKaRa), 0.3 μM of each primer, and approximately 10 ng of genomic DNA. The thermal profile consisted of an initial denaturation step of 94 °C for 2 min, followed by 30 cycles of denaturing at 98 °C for 10 s, annealing at 60 °C for 15 s, and extension at 68 °C for 8 min. The two amplicons from each WBPH individual were combined and sent to BIOZERON Company (Shanghai, China) for sequencing with the Illumina Hiseq 4000 platform. We sequenced 1 Gb of data for each mitogenome. Clean mitogenome sequence reads of each individual were assembled and annotated in Geneious 7.1.3 (Biomatters Ltd.Auckland, New Zealand) by using published WBPH mitogenome sequences (GenBank: NC_021417.1) as references. The assembly was performed using the “Highest Quality” option under the Consensus threshold setting, which mapped the highest quality base at each position onto the reference sequence. The minimum coverage for each base was 500×. Published SBPH mitochondrial genome data (GenBank: JX880068) were also used for analysis. The sequence of the 13 PCGs was extracted from the WBPH and SBPH mitogenomes. The sequences of these PCGs were aligned individually by codons using the MUSCLE method in MEGA 11.0.13 software [33] and concatenated for subsequent analyses.

### 4.4. WBPH Population Genetic Structure

To evaluate the genetic diversity of the WBPH populations, we calculated the number of haplotypes (N_h_), the number of private haplotypes (N_p_), haplotype diversity (H_d_), and nucleotide diversity (π) using DnaSP 6.12.03 [34]. The level of population differentiation was assessed by calculating pairwise *F*_ST_ values using DnaSP 6.12.03. To visually represent the genetic relationships among populations, a principal coordinates analysis (PCoA) was conducted using the pairwise *F*_ST_ matrix, and the resulting plot was generated using GraphPad Prism 9.5.1.

### 4.5. Mitogenome Genetic Diversity Comparison

The proportions of synonymous (Syn) and non-synonymous (Nsyn) substitutions for each PCG in the two planthopper species were calculated using DnaSP 6.12.03. Subsequently, we used Wilcoxon signed-rank tests to determine whether the proportions of Syn and Nsyn substitutions differed significantly between the two planthopper species.

### 4.6. dN/dS Analysis

To compare selective constraints acting on each mitochondrial PCG in each species, we calculated the dN/dS ratio. dN and dS were calculated using DnaSP 6.12.03 software. Because Syn mutation was lacking between some pairwise sequences, we calculated the ratio of dN/(dS + constant) as an alternative to dN/dS, where the constant was of the Ks for one Syn mutation, to avoid dividing by zero following the method of Mishmar et al. [35]. The Wilcoxon rank-sum test implemented in R 4.1.2 software was used to determine whether these values for each gene differed significantly between the two species.

### 4.7. McDonald–Kreitman Test Analysis

To identify genes that experienced positive selection, we performed the MK tests for each PCG. The MK tests were conducted by the web-based standard and generalized MK-test software (http://mkt.uab.es/mkt/MKT.asp) (accessed on 24 November 2023) [36] with Jukes and Cantor corrections [37]. *p* values were corrected for multiple comparisons through false discovery rates using the p. adjust function in R 4.1.2.

### 4.8. Genetic Distance

To evaluate the divergence between the mitogenomes of the two species, we calculated the genetic distances at each PCG under the Kimura 2-parameter model (K2P) with the bootstrap values of 500 in MEGA 11.0.13. The data were then visualized using GraphPad Prism 9.5.1.

## 5. Conclusions

The mitochondrial genomes of the two species appear to undergo predominantly purifying selection, indicating nearly neutral evolution. However, the 13 PCGs experience varied strength in purifying selection. Specifically, *ND1*, *ND2*, *ND6*, *COIII*, and *ATP8* exhibit significantly stronger purifying selection and greater divergence between the two species compared to the other genes. We also found that the mitochondrial genomes of WBPH undergo relatively stronger purifying selection than those of SBPH. However, we only compared two species in this study; more species (e.g., *Sogatella kolophon* (Kirkaldy, 1907) and *Nilaparvata lugens* (Stål)) are necessary to include to examine the influence of climate on the evolution of mitochondrial genomes. Furthermore, we suspect that mito-nuclear epistatic interactions probably play a role in maintaining nonsynonymous polymorphisms, particularly for *COI* and *COII*.

## Figures and Tables

**Figure 1 ijms-24-16847-f001:**
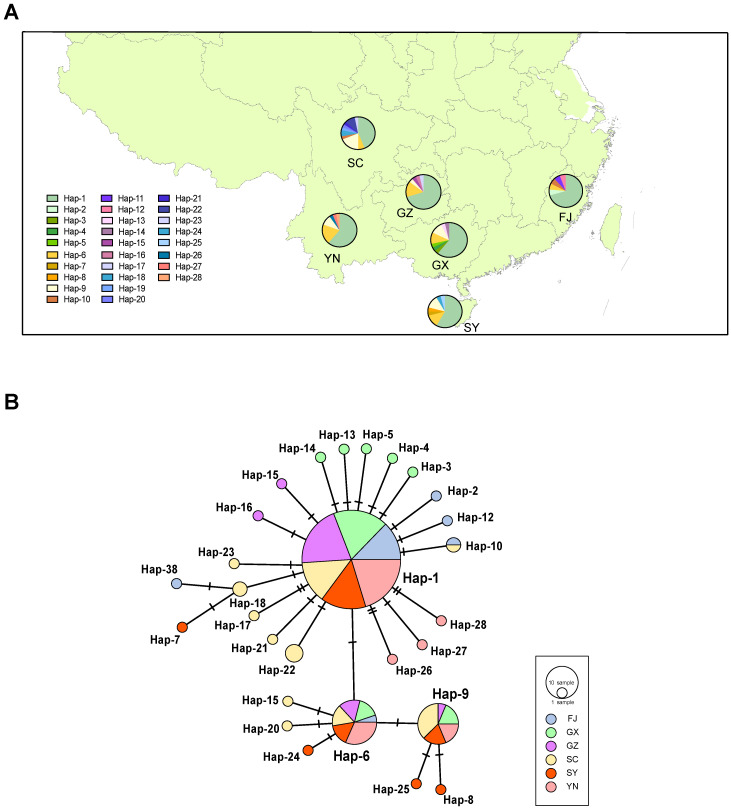
Sampling locations and geographical distribution of mtDNA haplotypes. (**A**) Sampling location and occurrence of mtDNA haplotypes in each location. (**B**) Haplotype network based on a partial mtDNA sequence (1849 bp), with haplotypes colored according to their geographical locations.

**Figure 2 ijms-24-16847-f002:**
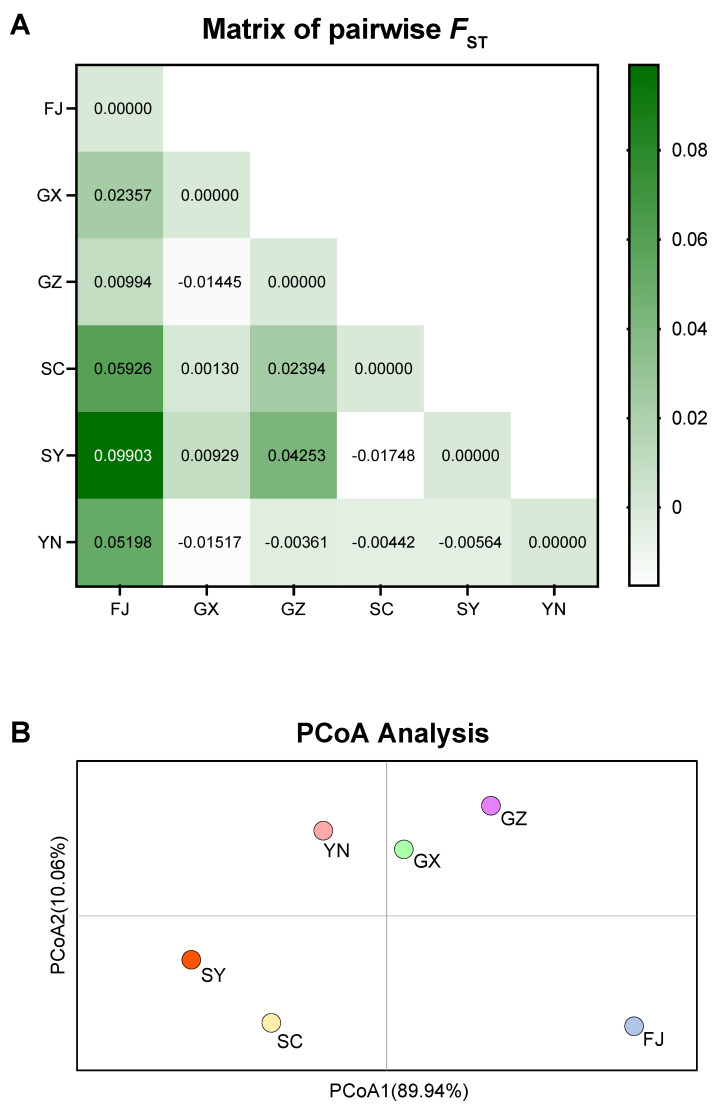
Heat map of pairwise *F*_ST_ values (**A**) and the corresponding principal coordinates analysis of the *F*_ST_ values (**B**).

**Figure 3 ijms-24-16847-f003:**
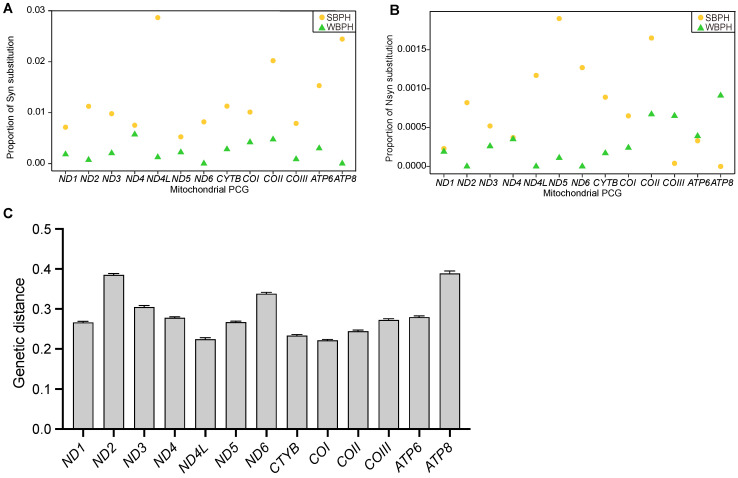
Proportion of (**A**) synonymous, (**B**) non-synonymous substitution in each mitochondrial PCG of SBPH and WBPH, and (**C**) K2P genetic distances in each PCG between the two species.

**Figure 4 ijms-24-16847-f004:**
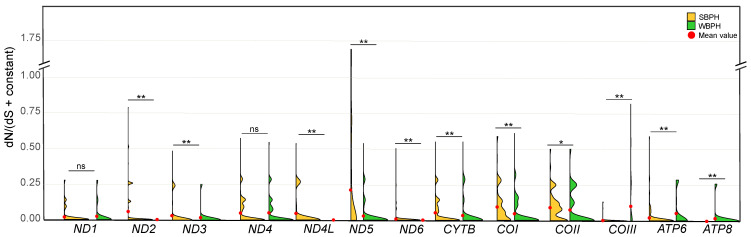
Comparison of relative constraints [dN/(dS + constant)] for each mitochondrial PCG between SBPH and WBPH. The Wilcoxon rank-sum test was used to determine whether these values for each gene differed significantly between the two species. ns, not significant; * *p* < 0.05; ** *p* < 0.01. The red dot indicates the mean value.

**Figure 5 ijms-24-16847-f005:**
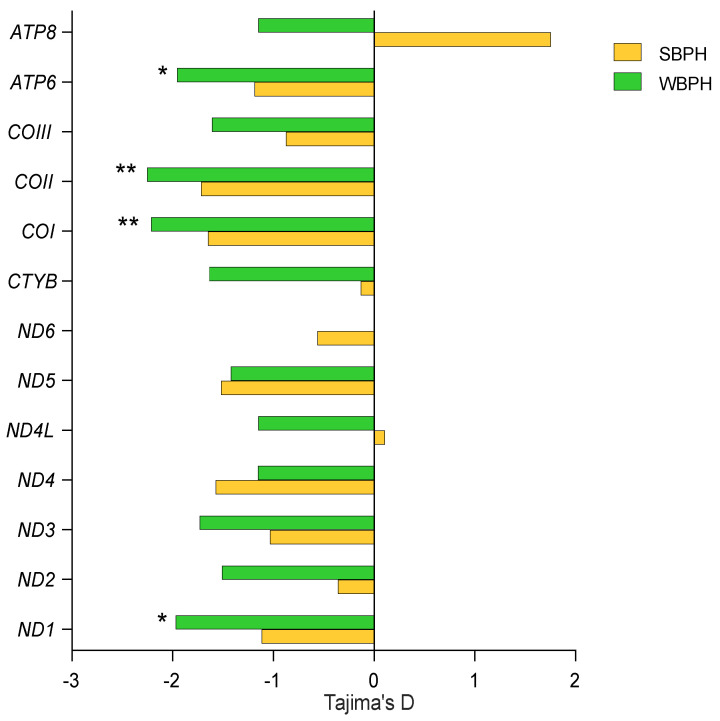
Tajima’s D tests for each mitochondrial PCG of SBPH and WBPH. * *p* < 0.05; ** *p* < 0.01.

**Table 1 ijms-24-16847-t001:** The genetic diversity of WBPH populations is based on an 1894 bp mtDNA sequence.

Locality/Zone	Number of Haplotypes, N_h_	Number of Private Haplotypes, N_p_	Haplotype Diversity, H_d_	Nucleotide Diversity, π(%)
FJ	6	3	0.51471	0.038
GX	8	5	0.62434	0.049
GZ	6	3	0.49573	0.037
SC	10	6	0.77621	0.074
SY	7	4	0.64855	0.068
YN	6	3	0.59355	0.050

**Table 2 ijms-24-16847-t002:** McDonald–Kreitman test with Jukes and Cantor corrections for each mitochondrial PCG.

Gene	Syn Divergence	Syn Polymorphism	Nsyn Divergence	Nsyn Polymorphism	NI	*p*-Value
*ND1*	270.02	16	78.44	3	0.645	0.492
*ND2*	361.84	12	166.08	6	1.089	0.866
*ND3*	68.56	6	51.94	4	0.879	0.848
*ND4*	196.47	24	197.25	14	0.581	0.118
*ND4L*	28.82	7	29.58	3	0.417	0.227
*ND5*	251.93	26	229.73	23	0.97	0.919
*ND6*	87.77	5	96.91	2	0.362	0.214
*CYTB*	387.62	16	52.04	9	4.189	<0.001
*COI*	518.98	50	40.93	13	3.296	<0.001
*COII*	250.13	30	26.64	20	4.55	<0.001
*COIII*	245.5	11	67.76	4	1.317	0.644
*ATP6*	144.28	22	71.26	5	0.46	0.124
*ATP8*	23.57	1	21.47	1	1.097	0.948

## Data Availability

The newly sequenced mitogenomes in the present study have been deposited in GenBank (OR491255- OR491282).

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
