# Peer review of "A Comparative Analysis of Selection Pressures Suffered by Mitochondrial Genomes in Two Planthopper Species with Divergent Climate Distributions"

_ijms, 2023, doi:10.3390/ijms242316847_

Round 1
Reviewer 1 Report
Comments and Suggestions for Authors
Authors compared the mitochondrial metagenomes of two Asian planthopper species from different climatic areas to see whether and how climate influences mitochondrial genome evolution. The results can be of practical importance because both planthopper species are serious rice pests in their geographical distribution area.
I must say that I am not an expert in mitochondria genomics, but I was impressed by the scientific quality of the manuscript. Even when some basis for the present study were already available from former papers of the group (e.g., the sequenced mitochondrial genome of one planthopper species), the present manuscript provides significant novel information. As may be expected, the results indicate a nearly neutral evolution of the mitochondrial genomes of the two species.
In my opinion, the paper needs only minor corrections before it can be accepted for publication:
- prelast line of p.1: insert a full stop after the bracket
- last line of Introduction: climate influences
- second and third line of 2.3: delete one "that"
- third line of Discussion: that produces
- last sentence of 4.1: delete the repetition in this sentence
- References must be checked for uniform spelling: paper titles in lowercase letters; journal names either in full or (correctly) abbreviated; species names in italics, etc.
Author Response
Many thanks for your helpful suggestions. We have revised the MS accordingly. The following is a point-by-point response to each of your comments and suggestions.
Question1: prelast line of p.1: insert a full stop after the bracket.
Response: Done.
Question 2: last line of Introduction: climate influences
Response: Done.
Question 3: second and third line of 2.3: delete one "that".
Response: Done.
Question 4: third line of Discussion: that produces.
Response: Done.
Question 5: last sentence of 4.1: delete the repetition in this sentence.
Response 5: Done. So sorry for this mistake.
Question 6: References must be checked for uniform spelling: paper titles in lowercase letters; journal names either in full or (correctly) abbreviated; species names in italics, etc.
Response: Thank you for your careful examination. We have thoroughly revised all the references.

Reviewer 2 Report
Comments and Suggestions for Authors
Dear Authors,
In my opinion, this publication has a speculative indication that these differences result from climatic differences. Climate involves a whole spectrum of other factors, other organisms, symbionts, etc. Although both species belong to the same tribe, they could have undergone these changes under completely different conditions, and only now they occupy different climatic zones, and the observed correlation of differences with the climate is accidental and does not result from selection pressure. Correlation does not mean causation. In my opinion, the entire article should clearly emphasize this (even more clearly than it already does because they themselves write that it is speculative) or, more broadly, base it on the literature of other known cases in the world of arthropods, not caenorhabditis. Another disadvantage of this work is that conclusions are drawn based on two species. In this aspect of scientific publications, other species (Delphacini) should be examined e.g. Sogatella kolophon (Kirkaldy, 1907) with Southeast Asia, not only important pest species on rice,
Regarding methodology, the publication contains standard research methods and the results are well-presented graphically.
Minor comments concern the ambiguity of some publication fragments marked in the PDF file.

Author Response
Thank you very much for your valuable comments and suggestions. We acknowledge that it is difficult in this work to draw a solid conclusion on the climate influence on the mitochondrial genome evolution, and other factors also play roles. This study represents a starting point for further examining the influence of climate on the mitochondrial genome evolution, which is a very challenging project. We agree with you that more species are necessary to include to get a solid conclusion in the future. We have been cautious during writing the conclusion in the previous version. In this version, we further turn down our tune in the introduction and conclusion for more conservation.
Question 1: (Horváth) author name should be added.
Response 1: Done.
Question 2: Check the sentence; direct of the moving species is not clear.
Response 2: The sentence has been revised as “It cannot overwinter in temperate Asian countries, including Japan, Korea and northern China, and migrates annually from subtropical regions (Indochina) to temperate regions (East Asia)”
Question 3: Fig, 1B presents the Haplotype network, but "each" species is not mentioned.
Response : Sorry for this mistake. All the haplotypes in Fig. 1B are of the white-backed planthopper. This sentence has been revised as “We characterized 28 haplotypes, with six to ten haplotypes for each population (Fig. 1B). The relationship among different haplotypes is shown in Fig. 1B.”
Question 4: Is it an error in the date, or were these samples taken so long ago?
Response: There is no problem with the sampling date. These samples were indeed collected in 2010.
Reviewer 3 Report
Comments and Suggestions for Authors
The study in the manuscript compares the mitochondrial genome in two planthopper species with divergent Climae distribution. The authors have interesting results that will be important for evolutionary biologists and scientists who work with climate change.
However, the paper requires careful editing: different Font sizes are used in many places.
In the Introduction, there is an overuse of etc.
Page 1 changes place for the period ... immunity, etc [6].
In the last paragraph, it would be helpful to add why authors have this aim...
Author Response
Many thanks for your helpful suggestions. The following is a point-by-point response to each of your comments and suggestions.
Question 1: However, the paper requires careful editing: different Font sizes are used in many places.
Response: Many thanks for your careful examination of our MS. We have carefully and thoroughly revised the front and size.
Question 2: In the Introduction, there is an overuse of etc
Response: There are 3 places where “etc” was used. But we think it is necessary to use there.
Question 3: Page 1 changes place for the period ... immunity, etc [6].
Response: Done.
Question 4: in the last paragraph, it would be helpful to add why authors have this aim...
Response: Sorry, I am not sure whether the last paragraph you mentioned is “5. Conclusion”. If so, we think it may lead to redundancy if we add the aim here. Because the aim of this study has been clearly interpreted in the last paragraph of “1. Introduction”.